# Visual SLAM for Unmanned Aerial Vehicles: Localization and Perception

**DOI:** 10.3390/s24102980

**Published:** 2024-05-08

**Authors:** Licong Zhuang, Xiaorong Zhong, Linjie Xu, Chunbao Tian, Wenshuai Yu

**Affiliations:** 1Guangdong Laboratory of Artificial Intelligence and Digital Economy (SZ), Yutang Street, Guangming District, Shenzhen 518132, China; 2210273099@email.szu.edu.cn (L.Z.); zhongxiaorong@gml.ac.cn (X.Z.); tianchunbao@gml.ac.cn (C.T.); 2The College of Civil and Transportation Engineering, Shenzhen University, 3688 Nanhai Avenue, Nanshan District, Shenzhen 518060, China; 2210474138@email.szu.edu.cn

**Keywords:** localization, perception, visual SLAM, UAV, odometry, feature extraction, visal–inertial SLAM, NeRF

## Abstract

Localization and perception play an important role as the basis of autonomous Unmanned Aerial Vehicle (UAV) applications, providing the internal state of movements and the external understanding of environments. Simultaneous Localization And Mapping (SLAM), one of the critical techniques for localization and perception, is facing technical upgrading, due to the development of embedded hardware, multi-sensor technology, and artificial intelligence. This survey aims at the development of visual SLAM and the basis of UAV applications. The solutions to critical problems for visual SLAM are shown by reviewing state-of-the-art and newly presented algorithms, providing the research progression and direction in three essential aspects: real-time performance, texture-less environments, and dynamic environments. Visual–inertial fusion and learning-based enhancement are discussed for UAV localization and perception to illustrate their role in UAV applications. Subsequently, the trend of UAV localization and perception is shown. The algorithm components, camera configuration, and data processing methods are also introduced to give comprehensive preliminaries. In this paper, we provide coverage of visual SLAM and its related technologies over the past decade, with a specific focus on their applications in autonomous UAV applications. We summarize the current research, reveal potential problems, and outline future trends from academic and engineering perspectives.

## 1. Introduction

UAVs have attracted much interest in their applications, such as unknown space perception, industrial defect inspection, and military operations, because of their flexibility, portability, and speed [1,2,3]. Localization and perception as the basis play an important role in those applications in which autonomous ability is needed. Internal states of movements and external understanding of environments are provided to enable UAV autonomous execution of missions. Localization determines whether the autonomous UAV moves accurately and acts precisely, while perception also supports basic movement in unknown space by detecting obstacles; moreover, a high-level understanding like semantic segmentation of environments enables more intelligent behavior, thereby enhancing the performance of autonomous UAVs and expanding the cover area of UAV applications. In particular, the above-mentioned applications have encountered the challenge of Global Navigation Satellite System (GNSS) denial where GNSS provides global localization capabilities. Therefore, an alternative approach to UAV localization and perception is needed, since it is the fundamental building block for autonomous upper missions such as navigation. The SLAM [4,5,6] technique has been widely researched for robot localization, and many excellent works have been presented. The SLAM technique is designed to simultaneously estimate the state (position, orientation) of the robot body and construct a map of the surrounding unknown environment through the data collected by the sensors. This technique requires not the GNSS signal but onboard sensors such as camera, Light Detection And Ranging (LiDAR), and sonar, and it provides robust, real-time localization and perception for autonomous robots, including UAVs.

Earlier SLAM techniques were intended to use LiDAR or multiple sensors to achieve accurate and robust localization [7]. The sensor configurations, however, required extensive cost and computation resources. With the advances in computer vision, visual SLAM techniques [8,9,10,11] take cameras as the only sensor input and have gained much popularity because of the image sensor’s low cost and simple configuration. Additionally, this sensor has great potential due to its ability to capture rich information about the surrounding environment. It is widely applied in lightweight devices such as smartphones, UAVs, and AR/VR equipment. This technique has a long history since [12] used the Kalman filter to estimate ego motion for the camera by extracting feature points in images in 1988. Nowadays, several well-designed and outstanding visual SLAM algorithms show incredible localization and mapping capability. To comprehensively understand the modern keyframe-based visual SLAM algorithm structure, we briefly introduce its workflow as following three main modules:Odometry: this is the basic module of the SLAM algorithm [13]. Its primary functionality is to process the latest received image by feature-based methods or direct methods, finding the correspondence between the current image and the reference image (or map). Once the correspondence is established, the camera pose can be estimated by epipolar geometry (2D–2D matches) [14], PnP (2D–3D matches) [15,16,17], or ICP (3D–3D matches) [18,19]. More recently, learning-based methods have been used for end-to-end estimating of the camera pose [20].Back end: this module maintains a global insistent map by performing Bundle Adjustment (BA) [21] for most state-of-the-art visual SLAM algorithms. On the one hand, the sliding window strategy is adopted, which keeps a fixed number of keyframes by marginalizing the old frame for controlling the BA cost in real time. On the other hand, a sparse map is constructed to optimize the motion and structure for more accurate results, since joint optimization with motion and dense map fail to run in real time.Loop closure: this module eliminates the accumulated error caused by large-scale, long-time estimation. To this end, a loop detection procedure is performed to detect the potential loop. Once a loop is detected, a lightweight pose graph optimization correlates with the trajectory, significantly improving the SLAM algorithm’s accuracy. Notably, the precision rate of the loop detection is critical and must be ensured. Otherwise, the wrong detection could directly lead the algorithm to fail.

With these three modules, a general visual SLAM pipeline can be depicted as in Figure 1, the real-time camera egomotion can be estimated, and the sparse (or dense) map can be reconstructed.

Returning to UAV localization and perception, compared to another widely used sensor, LiDAR, here are some reasons why visual SLAM algorithms are more suitable:Low-cost sensor: Visual sensors are cheap and low-power. This is important for UAVs, which have relatively unstable control, poor loadability, and low power consumption. Unstable control results in a high damage rate and destruction of sensors; poor loadability and low power consumption mean the weight and power consumption are better to be lower. These factors make visual sensors popular in UAV applications.High frame rate: Visual sensors can capture images at a higher frame rate, enabling algorithms to provide more localization information. To utilize the flexibility of UAV, high-frame-rate odometry is necessary for precision of control.Capturing rich texture information: This is beneficial for UAV perception; rich texture information brings a high understanding of environments, subsequently enabling more intelligent missions for UAVs such as object tracking, semantic segmentation, and implicit reconstruction.

Prior to our review, there have been excellent surveys on visual SLAM and related techniques, and it is necessary to summarize those surveys for comprehensive covering research: Ref. [25] reviews Visual Odometry (VO) and SLAM solutions for solving robot localization and mapping, introducing different types of VO methods and map representation, and EKF-based, particle filter-based, and RGB-D SLAM. It systematically explains algorithm, theory, and performance; Ref. [26] focuses on the types, approaches, challenges, and applications of VO. It shows the advantages of VO by comparing it with other sensor-based localization methods; Ref. [27] completely analyzes the SLAM problem and reveals SLAM’s capability and challenges. Moreover, it summarizes the SLAM systems over the past 30 years and discusses the long-term autonomy of SLAM algorithms, representation of mapping, theoretical tools, active SLAM, unconventional sensors, and learning methods, providing a comprehensive review and open problems. With the great advance of visual SLAM algorithms from 2010 to 2016, Ref. [7] summarizes the state-of-the-art algorithms, from conventional feature-based methods and direct methods to RGB-D camera-based methods, within this period; Ref. [8] reviews visual SLAM algorithms through their flowchart, providing a clear understanding of the advantages and shortcomings of the main algorithms in three aspects (visual only, visual–inertial, and RGB-D camera-based); Ref. [9] provides an in-depth literature survey on visual SLAM algorithms in various aspects including multi-sensor, feature type, environment, resource constraint, and odometry method. Discussing the current trends of this technique, Ref. [10] reviews the state-of-the-art visual SLAM algorithms to date—specifically, sensors, feature extraction and matching methods, deep learning techniques, and major datasets are mentioned. However, in our survey, we provide coverage of visual SLAM and its related technologies over the past decade, with a specific focus on their applications in autonomous UAV applications.

This survey focuses on the visual SLAM algorithm applied in UAV applications and discusses its trend in UAV applications. The paper structure is described as follows: Section 2 introduces sensor configurations and data processing methods for some preliminaries. In Section 3, we review the development of visual SLAM (or odometry) algorithms by features: real-time performance, texture-less environment, and dynamic environment emphasizing state-of-the-art algorithms. Additionally, Section 4 introduces the widely applied SLAM algorithms in UAVs called visual–inertial SLAM and illustrates how it improves the robustness and accuracy of UAV localization. Finally, learning-based modules for UAV perception are discussed in Section 5, and a comprehensive conclusion is drawn in Section 6.

## 2. Camera Configuration and Data Processing

The visual SLAM algorithm takes visual sensors, which are low-cost and have great potential, as the input. It can be varied by different camera configurations and data processing methods, which determine the algorithm inputs and lead to the successive modules changing. In this section, we introduce the different camera configurations and their properties, respectively, particularly for the feature-based method, the traditional feature extraction algorithms are reviewed, and at the end, the Inertial Measurement Unit (IMU) pre-integration method is briefly described to give the preliminary of the later section.

### 2.1. Camera Configuration

Different camera configurations significantly influence the performance and application scenes of visual SLAM algorithms. The advantages, shortcomings, and algorithms relative to sensor types are summarized in Table 1.

#### 2.1.1. Regular Camera Type

Monocular, stereo, and RGB-D cameras are the most common configurations for visual SLAM algorithms. The monocular system takes a single camera as the input. This cheap and straightforward sensor brings several challenges, and many researchers have dedicated themselves to overcoming them to outperform results with this simple sensor. One of the biggest challenges for the monocular system is scale ambiguity because it cannot measure the scene’s depth and estimate the up-to-scale motion; subsequently, the system inevitably suffers from scale drift, which can significantly reduce its accuracy. On the contrary, the stereo system can measure depth with a pair of cameras fixed with a constant baseline (or, more generally, with an overlapping field of view and fixed extrinsics). Subsequently, the disparity of the two cameras is computed by the stereo matching algorithm to produce the depth point with the actual scale. Notably, dense depth points can be made by epipolar searching. However, the quality is undermined when facing repetitive texture and poor illumination. Similarly, the RGB-D system can measure depth by active stereo or time-of-flight sensing. Therefore, a dense depth image can be directly produced without computation resources and does not rely on environment texture and illumination. However, a limitation exists, since the sunlight can firmly interrupt active sensing. This sensor is only suitable for indoor or short-distance depth measurements.

#### 2.1.2. Special Camera Type

In addition to these regular camera configurations, other practical configurations have emerged for more challenging environments. The event camera, a bio-inspired sensor, is sensitive to dynamics and intensity changes, captures every pixel asynchronously with low latency, and is suitable for dynamic object detection. As a new technique, it is barely researched, and there is still much potential to be unearthed. A multi-camera configuration can enlarge the Field of View (FOV) of the system, thus enabling the system to receive more information that resists the texture-less, dynamic environment and motion blur [40]. Furthermore, a more complete, well-distributed map can be constructed. For designing the multi-camera system, the multiple inputs must be carefully processed to extract the most useful information that enhances performance and maintains acceptable cost.

### 2.2. Data Processing

A critical step in the Visual Odometry algorithm is building the connection between the new incoming frame and the current estimation (referent frame or local map). Traditionally, there are two methods to do this: feature-based methods and direct methods. Feature-based methods require the feature extractor to extract features and match the features to find correspondences in the reference frame, as in Figure 2. Those methods, however, rely on the feature extractor to extract the invariant salient point, which is robust for rotation, view change, and illumination change. Additionally, there is always a trade-off between robustness and efficiency, while the robust extractor can increase the computation cost. On the contrary, direct methods discard the feature extractor and directly utilize the pixel intensity to align the new incoming frame by minimizing the photometric error. Notably, those methods use full-image information that subsequently achieves more accurate results and is naturally robust against poor texture environments and is efficient, since there is no need to extract features.

#### 2.2.1. Feature Extraction Algorithms

While the direct method seems to outperform the feature-based method, some disadvantages make the feature-based method worthwhile to develop continually. First of all, the direct method is based on the assumption that intensity invariance and scene illumination changes will be a disaster for this method. Secondly, this method has failed to build a strong association that limits the performance in long-term large-scale SLAM algorithms. Anyway, the feature-based method and feature extraction algorithms are still important. Feature extraction algorithms are widely researched in computer vision, and they traditionally detect features with pixels that are distinctive by a manually designed formulation. The Harris corner detector [42] is broadly used in computer vision tasks that extract the points by computing their intensity change in a small region. Features with rotation and illumination invariance can be extracted efficiently. The Tomasi corner detector [43] is similar to the Harris corner detector, with a proposed feature selection criterion. This detector extracts the good features that are more robust and less outlying, afterward improving the SLAM algorithms since the outliers could be harmful and interrupt the motion estimation.

However, the Harris and Shi–Tomasi corners lack scale invariance that cannot be matched in close or away motion. A Scale-Invariant Feature Transform (SIFT) algorithm was proposed by [44] to extract robust features to scale, including illumination, view changes, image rotation, and noise. This algorithm uses a coarse-to-fine approach to detect the features, which initially using the algorithm efficiently to identify potential features and refine them for solid invariance. It creates the descriptor for feature matching by computing the gradient magnitude and orientation in a designed region. However, the high computational cost hinders its usage in the visual SLAM algorithm, which requires real-time performance in limited computation devices. By simplifying the existing Hessian matrix-based detector and gradient distribution descriptor, the SURF algorithm [45] extracts the features with comparable repeatability, distinctiveness, and robustness against SIFT but improves efficiency. This algorithm uses Hessian matrix approximation to reduce the computation in the detector. It describes a distribution of Haar-wavelet responses in a 64-dimensional region of the feature, thus reducing the computation cost for detection, description, and matching.

Aiming at low-power CPU devices with limited computation and parallelizing ability, Rublee et al. proposed an efficient feature extraction algorithm called ORB [46]. Building on the FAST corner detector [47] and the BRIEF binary descriptor [48], they designed an efficient method to compute the orientation of FAST corners (Oriented FAST) and a rotation-aware BRIEF descriptor, improving performance while rotating. BRISK [49] is another efficient feature extraction algorithm that experimentally shows more scale invariance against ORB features with an acceptable computational cost. This algorithm adopts the scale space feature detection method inspired by AGAST [50] to improve scale invariance. It uses a compact binary string to describe the features similarly to BRIEF but with different sampling patterns. For those two algorithms, feature detection can quickly detect extensive features, and the efficient binary descriptor further improves the efficiency. This is contrary to the idea that the quality of features is better than the quantity, and it introduces the outliers into the visual SLAM algorithm. However, this is the trend of feature extraction algorithms in visual SLAM; robustness and accuracy are compromised for efficiency. This is because the outlier ejection method can later filter the extracted features to improve the performance of the SLAM algorithm, similar to the coarse-to-fine procedure that puts the main computation into outlier ejection, which deals with feature-level data to reduce overall computation cost, instead of feature extraction, which deals with pixel-level data. Of course, with acceptable efficiency depending on the devices and applications, the higher quality of features can still enhance visual SLAM algorithms.

KAZE [51] detects and describes features in a nonlinear scale space by means of nonlinear diffusion filtering to improve their repeatability and distinctiveness. AKAZE [52] improves the efficiency of the KAZE algorithm to make it available for embedded devices; it uses Fast Explicit Diffusion (FED) to dramatically accelerate feature detection in nonlinear scale spaces, and a Modified-Local Difference Binary (M-LDB) descriptor to efficiently describe features. to evaluate their detection, description, and matching performance. In [53], 14 feature extraction algorithms in 10 extremely variant image pairs were compared. A comprehensive comparison of the above-mentioned algorithms was presented by [54].

#### 2.2.2. IMU Pre-Integration

Specifically, the visual SLAM algorithm can be integrated with an Inertial Measurement Unit (IMU), usually containing an accelerometer and a gyroscope. The algorithm can be more accurate and can solve temporary visual tracking fails by leveraging the self-motion measurement provided by IMU. For those algorithms (visual–inertial SLAM or odometry), the IMU regularly measures the acceleration and angular velocity at a high rate, since the modern loosely coupled visual–inertial algorithm considers IMU measurements as variables in the back-end factor graph to perform optimization. However, problems arise. On the one hand, high-rate IMU measurements will dramatically increase the number of variables, increasing the optimization and computation cost scale. On the other hand, the optimizable variables are generated by IMU integration (acceleration is integrated as velocity, velocity is integrated as translation, and angular velocity is integrated as rotation): they contain the integration operation and, thus, are hard to optimize. To solve these problems, IMU pre-integration methods are proposed. In general, IMU pre-integration [55] summarizes IMU measurements between two concussive image frames to a single compound measurement, as Figure 3, which constrains the frame-to-frame motion and is easy to optimize.

According to the above illustrations, some essential preliminaries are given. The algorithm input and property can be determined by the sensor configuration and the scheme to process it. For instance, the monocular inertial SLAM algorithm, jointly visual and inertial initialization, can recover the real scale to avoid scale ambiguity. Furthermore, the IMU constraints are created by pre-integration to perform visual–inertial optimization, which refines the estimated scale. Especially for UAV applications, low-power, light-weight, and low-cost sensors are prior considerations due to the characteristics of limited power consumption and vulnerability. Additionally, heavy data processing cannot be afforded for embedded devices.

## 3. Visual SLAM Algorithms

Like Structure From Motion (SFM) [56], which estimates camera motion and constructs the unknown environment, the visual SLAM solves problems. However, they have a different emphasis: the SFM technique is a classical subject in the computer vision area; it reconstructs the 3D scene from a set of images (video stream or random images) and is allowed for offline computation. The visual SLAM technique was initially proposed for robotics applications and required real-time computation, emphasizing accurate, robust localization rather than mapping. In other words, the map served for localization. This section reviews the state-of-the-art or classic visual SLAM and odometry algorithms and three features: real-time performance, texture-less environment, and dynamic environment. We categorize those algorithms to illustrate the trend of development. We start from the early feature-based algorithms for real-time performance and show how to achieve real-time processing. At the same time, the essential requirement is satisfied, and the robustness and accuracy of the algorithms are considered, specifically in two typical environments: texture-less and dynamic.

### 3.1. Real-Time Performance

Global Bundle Adjustment (GBA) is widely used in SFM algorithms to optimize whole structures and poses jointly; it achieves accurate results but with too much computation cost. Therefore, this scheme was deemed unable to be applied in visual SLAM algorithms at an early stage. To this end, filtering schemes were adopted to solve the SLAM problem [57,58,59,60]. Among those filtering schemes, the Extended Kalman Filter (EKF) is the most widely used because it efficiently propagates the state and uncertainty. A top-down Bayesian framework to perform visual SLAM was proposed by [57]. In this framework, state estimation is computed by first-order uncertainty propagation in constant time. MonoSLAM [58] has successfully applied visual SLAM in interactive augmented reality and humanoid robots in room-sized domains with a single free camera. Using the EKF scheme, the state of the sensor can be incrementally estimated. However, this scheme can drift over time, since the state estimation only considers the last state, yet there are other past states. Furthermore, the computation cost increases along with new incoming features. These problems lead to filtering-based visual SLAM failing to deal with long-time large-scale scenes.

While filtering-based visual SLAM seems to reach its limitations, to overcome these limitations, Refs. [61,62] used the Smooth Variable Structure Filter (SVSF) to solve the SLAM problem. This filter is significantly robust against uncertain parameters and unknown noise characteristics. Both methods were shown to outperform conventional filtering-based methods in terms of accuracy and robustness. The Adaptive Smooth Variable Structure Filter (ASVSF) was proposed by [63], introducing a covariance matrix to assess the estimated uncertainty of the original SVSF and achieving more robust localization performance, especially in unstable noise disturbance. Furthermore, Ref. [64] deals with the dynamic environment by removing the dynamic information. Recently, Ref. [65] proposed a monocular–inertial SLAM algorithm based on SVSF, achieving a real-scale localization solution for UAV navigation. Overall, SVSF-based SLAM algorithms show great performance compared to conventional filtering methods and are capable of handling uncertainty and extensive noise.

Moreover, experiment [66] further shows that the BA scheme is more suitable for visual SLAM, in terms of accuracy, robustness, and efficiency. Local BA instead optimizes a batch of local keyframes and map points for real-time processing and has become the mainstream of current visual SLAM algorithm research. Along with advances in semiconductor technology, the CPU has become more and more parallel. To utilize this property, PTAM [67] is presented for tracking a hand-held camera in a small AR workspace by splitting the SLAM algorithm into tracking and mapping, respectively, running in two separate threads. In the tracking thread, a coarse-to-fine procedure is executed to estimate the current camera pose based on the feature-based method. At the same time, local and global BA are performed in the mapping thread to optimize the poses and sparse map points jointly. A keyframe selection strategy is adopted to control the optimization scale, which intensively reduces the amount and improves the quality of the optimizable variables. This algorithm shows the great advantage of the BA scheme against the EKF scheme. Furthermore, this parallel pipeline has been acknowledged and has significantly influenced the later visual SLAM algorithms. A dense visual SLAM algorithm for RGB-D cameras, similar to PTAM, was proposed by [32]. They split the algorithm into two components: fast odometry to register the current frame to the keyframe by direct method and a pose graph built by keyframe selection and optimized when a loop is detected. This algorithm uses a novel entropy-based keyframe selection strategy, inserting the frame when estimation uncertainty grows. However, the above two algorithms are only suitable for small-scale and indoor scenes. LSD-SLAM [28] can track camera motion using a monocular camera and can build a large-scale, semi-dense map in real time on a CPU device. This algorithm uses a filtering method to estimate the semi-dense depth map of keyframes, and the 3D similarity transforms between keyframes as edges for scale-aware global optimization. As we see, the feature-based, dense direct, semi-dense direct SLAM algorithms are designed by a multi-threads pipeline, and the keyframe selection strategy is adopted to select those essential frames and bridge the odometry to back-end optimization. Specifically, the real-time performance of the visual SLAM algorithm is determined by Visual Odometry, and the later optimized map provides a reference for odometry for more accurate results.

ORB-SLAM [22] is one of the most famous and classic visual SLAM algorithms; it further parallelizes the visual SLAM algorithm as three threads: tracking, local mapping, and loop closing. The local mapping thread provides intermediate results between initial tracking and final global optimization to build local data associations that efficiently optimize the tracking reference to enhance quality. The system is efficient, consistent, and reliable, using the same fast ORB features for all algorithm components. The new frame is tracked in the tracking thread by extracting the ORB features that match the local map. A fixed window is managed in the local mapping thread, and keyframes and map points within this window are optimized as the local map. In the loop closing thread, the loop is detected by DboW2 [68] and the global pose graph BA is performed to eliminate the accumulated drift. This algorithm further parallelizes the pipeline to create short-term, mid-term, and long-term associations, achieving state-of-the-art outdoor and indoor scene performance. ORB-SLAM2 [23] is an extension of the previous version; it takes multiple types of sensors as algorithm input, including monocular, stereo, and RGB-D cameras, and adds a new thread to perform global BA that jointly optimizes all keyframes and map points for more accurate results.

In conclusion, the multiple-thread visual SLAM pipeline is currently mainstream in this area. By splitting tracking and mapping, the real-time performance of visual SLAM systems is determined by the tracking thread. Thus, the above BA-based systems sacrifice accuracy in the tracking thread to ensure its efficiency. For instance, Refs. [22,67] use a fast and fairly robust feature extractor, Ref. [28] rather than using a semi-dense formulation to boost the tracking speed. Additionally, in filtering methods, real-time performance can easily be acquired, especially by redesigning the optimizable variables to reduce the propagation scale. However, this incremental propagation still suffers severely from accumulated drift. To this end, combining filtering methods and BA methods organically should be promising. To use filtering methods in the tracking thread, while in the mapping thread, BA methods refine the accuracy by integrating history information. In other words, filtering methods are more suitable as an odometry algorithm than a SLAM system. Ultimately, the real-time performance of visual SLAM algorithms seems to be enough for current desktop devices or even embedded devices by following this multiple-thread pipeline. However, it is necessary to further improve efficiency and save resources so that enabling more algorithms can be implemented. Performance is always compromised for efficiency, such as by simplifying the feature extractor or reducing the scale of optimization. Therefore, how to improve efficiency without sacrificing performance is crucial. Opinions for further improving efficiency from an engineering perspective are listed as follows:GPU boosting. A GPU has the capability of highly parallel computing and can be widely applied in learning-based methods. A common usage of GPUs is to boost feature extraction, since a GPU is a good computing matrix; however, in a visual SLAM system, there are other products that can be parallel-computed, such as map generation and solving BA equations. By utilizing the parallel performance of the CPU and the GPU, the efficiency of visual SLAM can be further improved.Data management. Data query is one of the most frequent operations; every new frame inputs to the system, finding correspondences to the reference frame, the local map, and the keyframe database. Therefore, how to use the appropriate data structure to manage the data stored in the system to achieve efficient queries is crucial. Especially in large-scale long-term visual SLAM systems.

### 3.2. Texture-Less Environment

White walls, space, and long tunnels, texture-less environments, severely undermine feature-based visual SLAM algorithm performance. The direct method performs more robustly in those environments and is accurate because of the utilization of all image pixels. It tracks the image by minimizing photometric error at the assumption of intensity invariance, subsequently saving the computational cost of feature extraction. This method can be categorized into three types: dense, semi-dense, and sparse (see Figure 4). The dense method is used in indoor scenes, with depth images provided by an RGB-D camera to construct the dense surface. The semi-dense approach builds the depth map of the vicinity of gradient pixels with the monocular camera and some prior geometry. However, both the dense and the semi-dense methods fail to jointly optimize pose and structure, leading to less accuracy. The sparse method constructs a sparse map of gradient pixels or patches, which can be jointly optimized with poses to achieve more accurate results.

#### 3.2.1. Dense Direct Formulation

For these dense methods, RGB-D cameras are usually required to produce dense depth images. KinectFusion [33] is presented for indoor real-time dense surface reconstruction; this algorithm tracks the camera pose by frame-to-model alignment, which ray-casts a global scene model to compute surface prediction and align the live frame through the multi-scale ICP method. Leveraging the GPU parallel computation, this algorithm integrates every frame measurement into the scene model, represented by the Truncated Signed Distance Function (TSDF). DTAM [69] uses a keyframe-based framework to construct a dense depth map by minimizing the global photometric error, requiring only an RGB sensor and commodity GPU hardware. In [32], the RGB-D camera is tracked by minimizing both photometric (intensity) and geometric (depth) errors. Also, a keyframe-based framework is used to construct the dense map. This algorithm does not need GPU enhancement but runs on the CPU device in real time. In [70], semi-dense monocular Visual Odometry is proposed, which continuously estimates a semi-dense inverse depth map of receiving frames. This algorithm represents the pixel inverse depth as a Gaussian probability distribution and propagates it frame-to-frame, constructing the vicinity of large intensity gradients of pixels. Additionally, it shows comparable performance against the dense method without depth cameras and GPU acceleration. LSD-SLAM [28] completes this odometry algorithm to the SLAM algorithm by maintaining a global map that contains a pose graph of keyframes with associated probabilistic semi-dense depth maps. The accumulated drift and scale drift are both reduced for larger-scale estimation.

#### 3.2.2. Sparse Direct Formulation

Since joint optimization of the pose and dense structure in real time is unaffordable, the map points are created initially and fixed with the associated frame, resulting in limited performance. SVO [71] uses the semi-direct method to track camera motion. It initially estimates the camera pose through sparse model-based image alignment, which minimizes the photometric error between pixel correspondences. Then, pose and structure joint optimization is performed through feature alignment by minimizing the reprojection error. In a separate thread, keyframe decision, feature extraction, and depth filter are used to construct the keyframe-based sparse map as a tracking reference. This algorithm utilizes the advantages of both direct and feature-based methods to achieve fast motion tracking. The latter extension [38] supports multiple camera configurations, edge tracking, and other camera models. With a full photometric calibration, including exposure time, lens vignetting, and non-linear response functions, DSO [29] proposes a sparse and direct formulation for Visual Odometry and shows superior performance against dense or direct methods. It minimizes the photometric error with the new formulation modeling the photometric parameters and jointly optimizes camera poses, affine brightness parameters, inverse depth values, and camera intrinsics. It shows the state-of-the-art performance and the great potential of direct sparse odometry.

#### 3.2.3. Structure Feature and Multi-Camera

While direct-based SLAM algorithms effectively solve texture-less environments, their limitations remain, as illustrated above. Structure feature-based algorithms extract points, lines, and planes in the environment to handle texture-less environments where the point features cannot be extracted. PL-SLAM [72,73] is built upon ORB-SLAM that extracts point and line features to track camera motion and perform optimization jointly. By formulating the representation of the line feature and its reprojection error term, those line features can be easily integrated with the original point feature-based algorithm. A new initialization approach is proposed based on only line correspondences that can estimate an initial map from three consecutive frames. Many excellent point and line feature-based algorithms that use the line features to improve robustness and accuracy, especially in texture-less environments, are presented [73,74,75,76,77,78]. In addition to the algorithm enhancement, texture-less problems can be solved intuitively by enlarging the camera’s FOV [79], fish-eye, omnidirectional [80,81], and multi-camera configurations [38,39,40]; they can receive more information about environments. To fully utilize those configurations, suitable camera models are considered, and relative strategies are proposed instead to migrate those configurations to existing algorithms.

In the context of a texture-less environment, through the comprehensive reviewing, two useful strategies to handle it can be summarized as follows:Fully utilizing current information, in such a situation, on how to extract useful information from images is crucial. Direct methods are proposed not only for discarding feature extraction but also to exploit the intensity information of images, making it possible to build correspondences with the current system. That is why direct methods also outperform feature-based methods. Unfortunately, intensity information is unstable compared with features and can be severely interfered with by photometric noise. Structure feature extracts useful information from another perspective. Line and plane are the common geometry similar to points and benefit for building correspondences. However, introducing these relatively complex geometries to the system undermines its efficiency and complexity. What is more, because of the limited FOV of the camera, the system will possibly face a structure-degeneration problem. On the one hand, direct methods need to reduce the influence of photometric noise such as photometric calibration and exposure control, or they need to combine with the feature extractor to provide more stable information, subsequently improving robustness. On the other hand, structure feature-based methods need to simplify the geometry expression and reduce system complexity to improve efficiency.Gathering more information, a texture-less environment usually means partial texture deficiency, because it is hard to find somewhere totally without texture. Therefore, simply enlarging the FOV of the sensor is a useful solution to gather more information and support to build correspondence. Subsequently, more data are inputted into the SLAM system, which will cause inefficiency and minimal improvement while in a rich texture environment. Therefore, reducing data redundancy, improving system efficiency, and building data connection (for multi-camera) is essential.

### 3.3. Dynamic Environment

We emphasize the static environment in the traditional visual SLAM problem. However, a real-world environment is always complex and dynamic. The moving objects presented in the image can interfere with the pose estimation and cause it to fail. Therefore, how to solve the SLAM problem in a dynamic environment has attracted attention in recent years, and is also the foundation of many applications. To solve this problem, several SLAM algorithms are presented and can be categorized into two types. The first type detects and removes the dynamic points or objects in front-end tracking; this method sees dynamic information as outliers, and it processes only static information to simplify the dynamic problem as a static problem, which is an easy and efficient way but fails to take advantage of dynamic information. The second type tracks the moving object while performing SLAM; this method tracks self-motion by constructing a map with a stationary background and moving objects that utilize the dynamic information and achieve more accuracy than removing them. Furthermore, object-oriented SLAM extracts the semantic information of the environment, including static objects and dynamic objects; joint optimization can be performed by building a consistent object-level map of the environment.

#### 3.3.1. Discarding Dynamic Information

RANdom SAmple Consensus (RANSAC) [82] is a popular method to remove outliers and improve system robustness. It randomly samples the data to fit the model containing the most significant number of inliers. PTAM, ORB-SLAM, and many visual SLAM algorithms have used this method to remove outliers, which keeps algorithms stable in slightly dynamic environments, which may fail when a large part of the image is dynamic. A prior-based adaptive RANSAC algorithm to handle the scene with many dynamic points was proposed by [83]. This algorithm is similar to the standard version but considers the distribution of inliers to fit the model accurately. While the above algorithms use RANSAC as the main scheme for outlier ejection, Ref. [84] proposed a depth edge-based RGB-D SLAM system. By weighting the points and edges to determine whether it is static or dynamic by creating the keyframe with a static feature, the frame-to-keyframe registration is performed for recovering the motion. A dense scene flow representation of the environment, to detect moving objects, was used by [85]. This algorithm performs coarse-to-fine estimation, first estimating the state in a regular way of odometry and later discarding the outliers to obtain more accurate results. For removing the object-level dynamic outlier, Ref. [86] used a Convolution Neural Network (CNN) to perform image segmentation with a priori dynamic objects. This algorithm builds upon the ORB-SLAM2 framework and segments the dynamic object using the Mask R-CNN module; furthermore, it can not only maintain a map with static points but also synthesize the frame without dynamic object occlusion by using a background inpainting module. These algorithms use the simplest way to perform SLAM in a dynamic world to remove the dynamic outlier, which is efficient and valuable. Still, if we utilized the dynamic information instead of discarding it, we could create a high-level understanding of the surrounding environment and even improve tracking accuracy.

#### 3.3.2. Utilizing Dynamic Information

For real-world applications of autonomous robots, a solution of Simultaneous Localization And Mapping (SLAM) and Moving Object Tracking (MOT) is desired, providing the fundamental function for high-level tasks such as autonomous driving and a higher understanding of the environment. A new discipline for this problem in theoretical and practical perspectives was established by [87]. Theoretically, it proposes a mathematical model to solve the SLAM and MOT problems jointly and builds a solid foundation. From a practical standpoint, it develops an algorithm to model perception, motion, and data association. SLAM++ [88] uses the ICP registration to track live images and detect 3D objects by leveraging prior knowledge through tracking 6DoF objects. An efficient pose graph optimization is performed with camera and object pose. DynamicFusion [89] is presented, which can track the non-rigid dynamic scene motion in real time using a single depth camera. This algorithm warps the scene geometry into the live frame to recover the scene motion without prior information. MaskFusion [90] is presented to track the multiple rigid objects in the scene by image-based instance-level semantic segmentation; this algorithm uses a mask network to update the new frame and then perform the motion tracking and object-level mapping. However, those two algorithms are designed for indoor scenes and still fail to utilize dynamic information to enhance the system’s performance. ClusterSLAM [91] proposes a back end for a stereo visual SLAM system that uses static and dynamic landmarks. By clustering the motions of dynamic rigid components, a decoupled factor graph optimization can be performed to estimate camera egomotion, static landmarks, and dynamic rigid motion. While the ClusterSLAM is only a back end that heavily relies on landmark tracking and association quality, ClusterVO [92] is proposed to contain a complete pipeline for either camera or moving object estimation. This algorithm extracts the ORB features and semantic bounding boxes and creates multi-level probabilistic association. For clustering the landmarks into rigid moving objects, the heterogeneous CRF module is used and, finally, state estimation is performed with sliding windows BA optimization. DynaSLAM2 [93] and VDO-SLAM [94] integrate the camera poses, static and dynamic points, and object motion into a BA factor graph optimization and utilize ynamic information, result in excellent performance.

In conclusion, discarding dynamic information is a simple but useful strategy to deal with dynamic problems and is especially suitable for indoor low dynamic scenes, since moving objects are random and relatively uncommon, simplifying problems from dynamic to static to maintain an efficient system. Utilizing dynamic information is more complex and suitable for outdoor scenes because pedestrians or cars usually have regular movements. However, whether discarding or utilizing dynamic information, localization is supported mainly by static information. Dynamic points filtering, semantic segmentation culling, object tracking, and scene flow tracking undermine and eliminate the disturbance of dynamic information. To this end, while static information is not enough, dynamic problems become texture-less problems. To deal with a highly dynamic environment, combining the direct method, structure feature, and FOV expansion is a practical scheme. Additionally, to enhance those SLAM systems coupled with a deep learning module such as semantic segmentation, the generalization, efficiency, and complexity of the model must be considered.

In this section, we first introduce the development of the visual SLAM algorithm structure, from filtering-based to optimization-based, from single-thread to multi-thread. This modern pipeline decouples localization and mapping from hard real-time constraints and changes the real-time requirement of the SLAM algorithm to the odometry algorithm. This parallelizing pipeline allows researchers to study and modify, helping engineering applications and academic research. We discuss two real-world problems and how the SLAM algorithm deals with a texture-less and dynamic environment. In texture-less environments, we emphasize direct-based algorithms, one of the two main branches of visual SLAM. Dense, semi-dense, and sparse methods show the development of this method. Subsequently, other methods that structure feature extraction and FOV expansion are briefly discussed. Two strategies are adopted in dynamic environments: discarding or tracking the dynamic points. The former is a simple scheme to handle dynamic environments, turning a dynamic problem into a standard static problem. The latter utilizes dynamic information to improve performance, incorporating optical flow, MOT, and semantic segmentation techniques. For both academic and engineering purposes, we list several open source visual SLAM algorithms and summarize their sensor inputs and features in Table 2.

## 4. Visual–Inertial Fusion for UAV Localization

The pure visual SLAM algorithm obtains information from the external surrounding environments and fails to sense self-motion. Environmental conditions (over-exposure, dusty conditions, and dark regions) can directly lead to deadly error in the algorithm. Therefore, IMU is needed to significantly reduce the influence of environmental conditions, especially in UAV applications [95,96,97], where robust localization is required to prevent accidents, such as losing control or dropping from the sky. Since self-motion can be attained and integrated into the visual SLAM algorithm, there are several advantages to integrating visual and inertial measurements:Inertial measurements provide extra constraints for pure visual back-end optimization and improve its accuracy and robustness.The real-world scale can be recovered for monocular SLAM algorithms to solve the scale ambiguity problem.The high-frame-rate inertial measurements can fast propagate the odometry information for autonomous robot agents.While visual tracking fails to maintain the odometry, inertial measurements could provide a temporary prediction for keeping the system working.

At the same time, integrating inertial measurement brings new challenges for the visual algorithm. On one hand, the high-frame-rate inertial measurement needs to be processed appropriately to fit the low-frame-rate visual system. On the other hand, the accumulated bias and error of IMU measurement need to be fixed by utilizing the visual information. For UAV applications, visual–inertial solutions have gained plenty of interest, and this sensor configuration significantly improves the algorithm performance in a relatively simple way against complex structure design, algorithm optimization, and mathematical formulation.

### 4.1. EKF-Based Visual–Inertial SLAM

Similar to pure visual SLAM, this technique also starts with filtering-based algorithms. In [98], an EKF-based visual inertial odometry uses static features to constrain inertial propagation. The algorithm is computationally efficient and can precisely estimate large-scale real-world environments. ROVIO [99] is proposed to use the photometric error of the multi-level patch as an innovation term in EKF propagation. By parametrizing features, the filtering operations can be applied. Overall, this algorithm propagates robot-centric rotation, translation, velocity, the transformation of IMU and camera, IMU bias, and feature parameters, and employs QR-decomposition to maintain computational efficiency. These EKF-based visual–inertial algorithms have experimentally demonstrated robustness and accuracy by integrating inertial and visual measurements. However, the optimization-based system could perform better in terms of robustness and accuracy by constructing a factor graph for joint optimization.

### 4.2. BA-Based Visual–Inertial SLAM

In [100], a tightly coupled visual–inertial SLAM was proposed, introducing the IMU error term integrated with feature reprojection error for joint optimization. A marginalization scheme was employed to maintain the visual constraints of keyframes for bounding computation complexity. ORB-SLAM-VI [101] proposes zero-drift localization by re-using the map; this algorithm performs optimization within a local window but considers a fixed window connected by a co-visibility graph and loop closure with a pose graph. In addition to this, a novel IMU initialization method is proposed that computes scale, gravity direction, velocity, and biases.

VINS-MONO [102] is a tightly coupled monocular inertial system. In [102], they propose a robust initialization procedure that performs vision-only initialization and then aligns metric IMU pre-integration with the visual-only result to recover scale, gravity, velocity, and biases. For front-end tracking, the system tracks the existing features by the KLT sparse optical flow algorithm and detects new features to maintain a minimum number of features in the current frame; simultaneously, IMU measurements within two frames are pre-integrated. If the system is initialized, a tightly coupled optimization includes pre-integration terms, features, and poses in a sliding window. Furthermore, they adopt the DBoW2 to detect and close the loop by 4-DOF global pose graph optimization, since IMU can provide absolute pitch and roll observation.

ORB-SLAM3 [24] further extended ORB-SLAM—which supports both visual and visual–inertial sensor configurations, including monocular, monocular inertial, stereo inertial, etc.—and introduced Atlas to save a set of disconnected maps that can be used for loop detection and relocalization, and which merge smoothly with the current connected map. This algorithm is based on ORB-SLAM2 and integrates the IMU measurements; it initializes the system in three steps: visual-only, inertial-only, and visual–inertial joint initialization. In the tracking thread, the algorithm continuously pre-integrates IMU measurements and estimates the pose by feature extraction and matching. When the tracking is lost, the system will attempt to predict motion by IMU measurements, and if this does not work, the relocalization mode will be executed. In the mapping thread, the IMU constraints will be added to the graph for joint optimization and will perform IMU scale refinement. In the loop closure thread, while the system detects a loop in ATLAS, two maps will be merged into one map, and essential graph optimization will be performed. This algorithm is complete, supports almost all visual sensor configurations and different camera models, and contains short-term, mid-term, and long-term data associations; therefore, it shows excellent accuracy and robustness.

VI-DSO [103] and DM-VIO [104] are monocular visual–inertial odometry, which minimize the photometric errors of sparse pixels with high-intensity gradient and IMU measurement errors. VI-DSO introduces a novel marginalization procedure called dynamic marginalization that maintains several marginalization priors to adapt the scale estimation dynamically. Preventing the scale is fixed by the marginalization prior, while it needs to be better estimated. DM-VIO proposes delayed marginalization to solve marginalization that is hard to reverse. This approach can inject the IMU information after initialization to the pure visual prior and replace the prior. At the same time, the scale estimate changes in the same way as VI-DSO. Additionally, a weighted photometric BA is proposed to adjust the weight of visual residuals dynamically. Those algorithms improve the marginalization procedure to better compute the priors in BA optimization. DM-VIO exceeds the state-of-the-art visual–inertial stereo algorithms and has shown its effectiveness. Table 3 summarizes the above algorithms with three basic components.

In conclusion, this technique builds upon visual SLAM and achieves excellent performance, in terms of robustness and accuracy, by integrating IMU measurements. Meanwhile, in UAV applications such as unknown-space perception, industrial-defect inspection, and military operations, this technique provides a suitable solution for localization. Specifically, for navigation, robust localization enables the UAV to fly in challenging environments with low illumination, over-exposure, and poor texture. For exploration and reconstruction, accurate localization provides a solid foundation to improve the quality and consistency of reconstruction. For flying control, more body states such as velocity and accelerated velocity and high-rate odometry information propagated by IMU support aggressive control to utilize the flexibility of UAV.

## 5. Learning-Based Enhancement for UAV Perception

In UAV applications, obstacle avoidance, path planning, and real-time reconstruction, a dense or semi-dense map will be required. However, for robustness, accuracy, and efficiency of localization, visual SLAM algorithms usually retain a sparse map for joint optimization. To this end, another denser map is constructed by obtaining depth images and associated odometry. Depth images are projected using a camera model to produce a dense point cloud, which is aligned by the odometry of the SLAM algorithm [105,106]. This approach usually requires depth sensors, such as Realsense D435i, Realsense D455, and Kinect v2, to provide depth images. This approach is limited by the performance of depth sensors like RGB-D cameras, which will be disrupted by sunlight and will fail to measure the depth of outdoor long-range scenes. Therefore, gathering the depth from original visual SLAM inputs gains excellent interest, reducing the sensors’ cost and improving the synchronization between depth image and odometry, since the depth sensor probably needs to be better synchronized with SLAM algorithm inputs. With the advancement of GPUs, Jetson developer kits such as Jetson Xavier NX, Jetson TX2, and Jetson Orin integrate GPUs into embedded devices to enable the implementation of learning-based modules. These low-cost devices are suitable for autonomous mobile robots such as UAVs and have been successfully applied in commodity drones such as Skydio. There are some advantages of learning-based enhancement for UAV perception:Learning-based methods are good for extracting texture information in images, and usually perform better than traditional methods.Learning-based perception uses the same inputs as in localization, improving consistency and, at the same time, saving the cost of sensors.With the rapid development of GPUs and Artificial Intelligence, learning-based methods have become mainstream.

Similar to embedded CPUs, embedded GPUs also suffer from power limitations that require the implemented learning-based module to be simple and efficient.

### 5.1. Monocular Depth Estimation

Monocular depth estimation, a technique to estimate the depth of pixels in 2D images, is significantly enhanced by deep learning. CNNs can extract richer and more complex feature representations than traditional approaches, which usually rely on hand-crafted features, scene assumption, and manual parameters adaption, subsequently achieving better results for depth estimation. In [107], two CNN stacks were used for monocular depth estimation: one stack to estimate the global scale depth, and another to refine the local detail. In [108], the same multi-scale architecture as in [107] was adopted to predict depth, surface normals and semantic labels. In [109], a unified deep CNN framework was used to learn the potential of a continuous Conditional Random Field (CRF), estimating the depth of general scenes without geometric priors and extra information. On the contrary, Ref. [110] did not rely on refinement or CRF, and proposed a full CNN architecture to estimate depth. This architecture was built upon ResNet, and it outperformed previous methods. In [111,112], the sparse pixel depth provided by the visual SLAM algorithm was used to improve the accuracy and reliability of depth estimation. Monocular depth estimation algorithms rely on deep learning-based methods that focus on estimation accuracy but with increasing computational complexity. To this end, FastDepth [113] proposed a lightweight encoder–decoder network to efficiently estimate the monocular depth map, which achieves comparable accuracy with embedded GPU devices in real time. In [114], a proper trade-off was achieved between accuracy and efficiency, assembling two encoder–decoder subnetworks to solve spatial information loss caused by the feature extractor. These efficient networks provide an alternative solution for UAV depth sensing, which gets rid of the limitations of depth sensors. Incorporating accurate localization, a promising dense depth map can be constructed.

### 5.2. NeRF-Based SLAM

Neural implicit fields (NeRF) [115] are novel representations that reconstruct high-fidelity surfaces and arbitrary view renderings of scenes. Compared to traditional point cloud reconstruction, NeRF-based reconstruction produces realistic illumination and high-quality images, resulting in excellent performance of complex scenes and detailed reconstruction. Visual SLAM incorporated with NeRF for camera tracking and dense reconstruction has recently been investigated and called NeRF-based SLAM. Unlike the traditional dense visual SLAM, this technique overcomes the drawback of failing to jointly optimize the structure and poses since this implicit representation is differentiable. The first NeRF-based SLAM to track an RGB-D camera pose in real time and to jointly optimize poses with a dense map was iMAP [116]. Following the traditional visual SLAM pipeline, tracking and mapping are run in two parallelizing threads to decouple the hard real-time constraints. NICE-SLAM [117] further improves the efficiency of tracking and mapping, incorporating multi-level local information. This algorithm is more scalable and robust than other NeRF-based SLAM algorithms. NeRF-based SLAM is a new trend in visual SLAM research. Therefore, some problems still need solutions. Urgently, the efficiency of algorithms needs to be improved to reduce the hardware requirement, which is usually a desktop setup. More recently, 3D Gaussian splatting [118] has been used for real-time radiance field rendering. By improving efficiency, these algorithms can be applied in UAVs, which will be a milestone for UAV reconstruction. In addition, current NeRF-based SLAM algorithms only support small-scale indoor scenes, with a solution still needed for large-scale outdoor scenes.

Regular pipelines of the above three schemes are depicted in Figure 5. In conclusion, along with the development of GPU-based hardware, the accurate and excellent performance of learning-based techniques for UAV perception have shown us new solutions. Lower sensor complexity and better performance make UAVs more simple and powerful. However, due to the limited computational source of embedded devices, the main problem or research direction for these learning-based modules to be applied in UAVs is the efficiency of algorithms.

## 6. Conclusions

This paper reviewed the development of visual SLAM in three aspects: real-time performance, texture-less environments, and dynamic environments. Introducing state-of-the-art and recently presented algorithms, we illustrated the mainstream of pipeline design and the two main applied environment problems. Furthermore, localization and perception for UAV applications were discussed, based on visual SLAM. For localization, we emphasized the widely applied visual–inertial SLAM to show how inertial measurements improve UAV localization and subsequently improve other tasks such as exploration. For perception, we demonstrated the capabilities of the newly presented learning-based methods. GPU integrated into embedded devices can make implementing learning-based modules for UAV perception possible.

Through decades of development, visual SLAM has become more complete and powerful, providing outstanding localization and mapping for various robotic applications. Recent rapidly developed learning-based approaches have continued to improve this technique, to transform traditional hand-crafted algorithms into data-driven algorithms, replacing conventional feature extraction, odometry, and even the whole system. However, while these data-driven algorithms perform better than hand-crafted ones, their generalization ability, interpretability, and efficiency are still open to question. Meanwhile, traditional visual SLAM modules can be improved for more challenging environments, hardware, and motion. To provide a strong foundation for UAV applications, multi-sensor fusion is an efficient scheme for localization and perception. Among these configurations, visual–inertial fusion is undoubtedly a simple and effective way to improve localization. Multi-camera fusion also improves perception and localization by enlarging the FOV. For multiple data inputs, efficiently processing them could be a crucial problem that filters redundant or wrong information and uses essential information to enhance performance. Additionally, the later keyframe decision, BA optimization, and map management need to be considered to adjust multiple inputs so that they can be fully utilized.

To handle complex scenes and missions, autonomous UAVs are required to be more reliable and intelligent. Therefore, robust localization and intelligentized perception are essential trends in autonomous UAV applications. Further explanations are discussed as follows:Robust localization: UAV localization may not be so accurate but is robust, especially for applications that require the UAV to cross various scenes, such as unknown space exploration. As in military investigation, dusty disturbance, complex environments, and the requirement for fast movement severely interfere with the data inputs. In cave or tunnel exploration, partial darkness, and reduced environmental texture also weaken information support for localization. These potential factors bring challenges to UAV localization, and only robust UAV localization can deal with various challenging environments, meeting the needs of the day-by-day growing complexity of applications.Intelligentized perception: A regular dense point cloud map or grid map is built for point-driven navigation; however, it is not adequate to support more and more intelligent missions, such as object searching in an unknown space. For instance, in disaster rescue, the regular map for navigation only supports the UAV to mechanically search victims, and it is inefficient. Intelligentized perception means a high-level understanding of surrounding environments: with this understanding, the UAV can infer potential victims by obtaining environmental clues, such as blood or a piece of clothing. Moreover, manual intervention can be significantly reduced and UAVs can take charge of strategy decisions, parameter adjustment, and risk prevention.

Additionally, there is the popular concept of lifelong SLAM [120] with continuous localization and mapping in the long term. Robust localization meets the needs of long-term localization in complex and changing environments. In addition, intelligentized perception helps the system to understand changes in scenes and objects for long-term mapping. Moreover, this understanding supports the system in mission planning and decisions.

This paper summarizes the research, including visual SLAM, visual–inertial SLAM, and learning-based SLAM, from different aspects, to comprehensively understand this technique. Furthermore, we have discussed the problems, advantages, and future trends of these relative approaches. With the advent of Artificial Intelligence, it is worth reviewing these conventional approaches, to ascertain potential and understand the foundation of this technique.

## Figures and Tables

**Figure 1 sensors-24-02980-f001:**
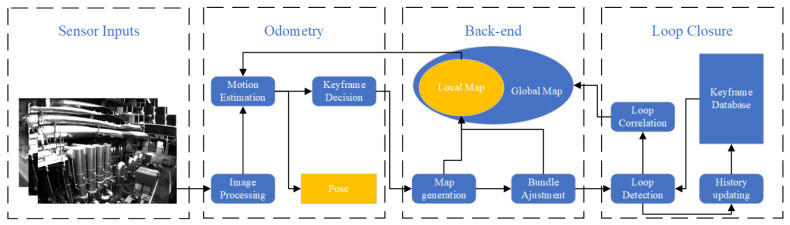
General keyframe-based visual SLAM algorithm pipeline (similar to [22,23,24]). ‘Image Processing’ includes image distortion, feature extraction, etc., and procedures for image matching to find correspondence; ’Keyframe Decision’ considers whether the current map supports odometry to estimate the current state; ’Map generation’ triangulates the pixels to 3D map points; ’History updating’ updates the dataset by inserting new keyframes for potential loop detection.

**Figure 2 sensors-24-02980-f002:**
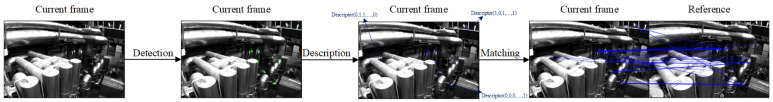
Finding correspondences between the current frame and the reference for the feature-based method (images come from EuRoC [41] dataset). Detection detects pixels with distinctiveness and repeatability; description creates the unique descriptor of features for feature matching; matching compares the similarity of those descriptors to match features.

**Figure 3 sensors-24-02980-f003:**
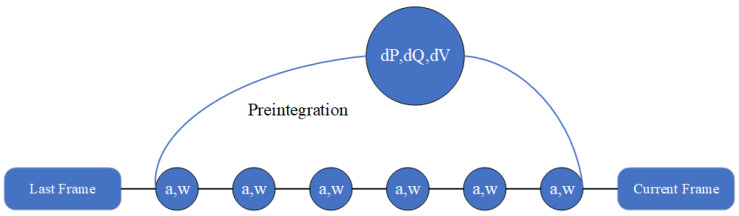
IMU measurements pre-integration: letters a and w, respectively, represent acceleration and angular velocity; dP, dQ, and dV are increments of position, orientation, and velocity between the last frame to the current frame.

**Figure 4 sensors-24-02980-f004:**
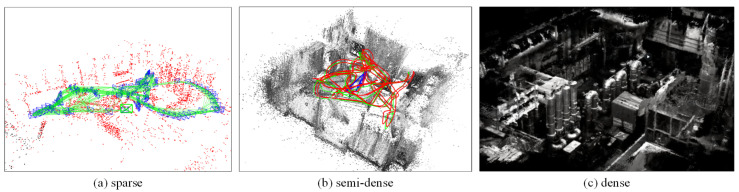
Different type of maps from EuRoC [41] dataset.

**Figure 5 sensors-24-02980-f005:**
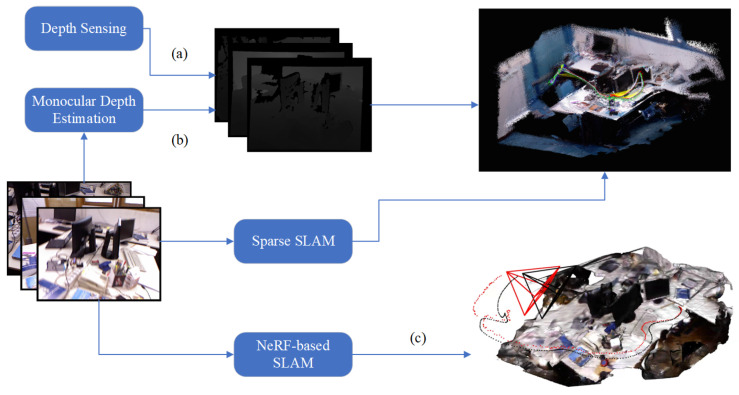
Three schemes to build a dense map (experiment dataset [119]): (a) and (b) use different ways to produce depth image, the former measuring depth by depth sensors, while the latter estimates depth by deep learning modules and reconstructs a dense map by retrieving the pose of the depth images; (c) a NeRF-based SLAM produces a NeRF representation of environments. Other dense SLAM algorithms usually result in poor performance because of the lack of joint optimization.

**Table 1 sensors-24-02980-t001:** Advantages, shortcomings, and algorithms relative to sensor types.

Type	Advantage	Shortcoming	Algorithms
Monocular	simple and cheap	suffers from scale ambiguity	[22,28,29]
Stereo	indoor and outdoor depth sensing	relies on texture of environment and requires computing	[23,30,31]
RGB-D	high-quality depth sensing	only suitable for indoor scenes and has high power consumption	[32,33,34]
Event	sensitive to dynamic information	unable to capture regular-intensity images	[35,36,37]
Multi-camera	captures more information and arbitrary views combination	larger data need to be processed	[38,39,40]

**Table 2 sensors-24-02980-t002:** Open source visual SLAM algorithms.

Odometry Method	Algorithm	Sensor	Feature
Feature-based	Mono-SLAM [58]	Monocular	EKF-based
	PTAM [67]	Monocular	parallel tracking and mapping
	ORB-SLAM [22]	Monocular	multi-threads
	DynamicFusion [89]	RGB-D	non-rigid dynamic scene motion tracking
	ORB-SLAM2 [23]	Monocular, stereo, RGB-D	multi-configurations
	PL-SLAM [72,73]	Monocular, stereo	point, line feature extraction
	Dyna-SLAM [86]	Monocular, stereo, RGB-D	segment dynamic objects
	MaskFusion [90]	RGB-D	tracks multiple objects
	VDO-SLAM [94]	RGB-D	joint optimization including camera poses, static, dynamic points, and object motion
	PLP-SLAM [78]	Monocular, stereo, RGB-D	point, line, plane feature extraction
Direct	KinectFusion [33]	RGB-D	dense surface reconstruction
	DTAM [69]	Monocular	monocular dense tracking and mapping
	DVO-SLAM [32]	RGB-D	tracks motion by minimizing both photometric and geometric errors
	LSD-SLAM [28]	Monocular	large-scale estimation
	SVO [38,71] *	Monocular, Multiple camera	hybrid odometry
	DSO [29]	Monocular	sparse direct formulation
	BAD-SLAM [34]	RGB-D	fast direct BA formulation

* SVO is a semi-direct odometry, using both photometric and geometric error.

**Table 3 sensors-24-02980-t003:** Open source visual–inertial SLAM algorithms.

Algorithm	Odometry Method	Optimization	Loop Closure
MSCKF [98]	feature-based	EKF-based	-
OKVIS [100]	feature-based	Local BA	-
ROVIO [99]	direct	EKF-based	-
ORB-SLAM-VI [101]	feature-based	Local BA	PGBA
VINS-Mono [102]	feature-based	Local BA	PGBA
VI-DSO [103]	direct	Local BA	-
ORB-SLAM3 [24]	feature-based	Local BA, GBA	PGBA
DM-VIO [104]	direct	Local BA	-

## Data Availability

No new data were created or analyzed in this study. Data sharing is not applicable to this article.

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
