# Peer review of "Visual SLAM for Unmanned Aerial Vehicles: Localization and Perception"

_sensors, 2024, doi:10.3390/s24102980_

Round 1

Reviewer 1 Report

Comments and Suggestions for Authors

This article, "Visual SLAM for Unmanned Aerial Vehicles: Localization and Perception," provides a comprehensive overview of the development and current state of visual SLAM technology, particularly in the context of UAV applications. It highlights the importance of localization and perception in autonomous UAVs and emphasizes the ongoing advancements in visual SLAM driven by embedded hardware, multi-sensor technology, and artificial intelligence. It presents the survey's aim to explore the development of visual SLAM in UAV applications, addressing critical challenges such as real-time performance, texture-less environments, and dynamic environments. The inclusion of visual-inertial fusion and learning-based enhancement demonstrates the evolving role of these techniques in UAV localization and perception. The conclusion summarizes the paper's findings, discussing the progress made in real-time performance, addressing challenges in various environments, and the integration of learning-based approaches.

Critically, while the paper provides a thorough overview of existing methodologies and emerging trends, it lacks in-depth analysis and critical evaluation of the reviewed approaches. Furthermore, while it discusses the importance of robust localization and intelligent perception for autonomous UAV applications, it could benefit from more concrete examples and case studies to illustrate these concepts in practice. Additionally, the paper could have delved deeper into the limitations and challenges associated with current visual SLAM techniques, as well as potential areas for future research and development. Moreover, the related work section could be expanded to include references to studies focusing on robust SLAM techniques, such as those based on the Smooth Variable Structure Filter (SVSF) and adaptive SVSF, for both static and dynamic environments. By incorporating a more comprehensive review of related literature, the paper could provide readers with a broader understanding of the existing landscape of SLAM research and its implications for UAV applications.

Comments on the Quality of English Language

The quality of English language in the article is quite good. The writing is clear, coherent, and generally free from grammatical errors. The abstract, conclusion, and other sections are well-structured and easy to follow, which enhances readability.

Author Response

Thank you very much for your time and effort to review our manuscript. We appreciate your precious suggestions and please find our responses below.

Comments 1:  While the paper provides a thorough overview of existing methodologies and emerging trends, it lacks in-depth analysis and critical evaluation of the reviewed approaches.

Response 1: Our manuscript does lack in-depth analysis and critical evaluation of the reviewed approaches. This is because our main purpose is to illustrate the development of the specific visual SLAM problem, so we emphasize the key influence of reviewed approaches related to problems such as real-time performance. However, it is a fact that readers can not completely learn the features of reviewed approaches. To this end, we summarize other excellent surveys to cover the reviewed approaches and provide a way to get the analysis and evaluation of reviewed approaches. We hope this can compensate for this drawback. Please find the corresponding revisions in Section Introduction (lines 91-111).

Comments 2: While it discusses the importance of robust localization and intelligent perception for autonomous UAV applications, it could benefit from more concrete examples and case studies to illustrate these concepts in practice.

Response 2: Agree. Concepts are unconvincing without concrete examples, additionally, examples would help to better understand concepts. To this end, we add some examples to illustrate why robust localization and intelligent perception are important. Please find our revisions highlighted in Section Conclusion.

Comments 3: The paper could have delved deeper into the limitations and challenges associated with current visual SLAM techniques, as well as potential areas for future research and development.

Response 3: Agree. It's important to provide our opinions on visual SLAM techniques by reviewing related research. However, our manuscript lacks discussion about current challenges and future development in Section 3. Therefore, we revise our manuscript in Section 3 to provide our opinions about the capabilities, limitations, and future trends in each aspect (at the end of subsections). Please find the corresponding revisions highlighted in the re-submitted files.

Comments 4: The related work section could be expanded to include references to studies focusing on robust SLAM techniques, such as those based on the Smooth Variable Structure Filter (SVSF) and adaptive SVSF, for both static and dynamic environments. By incorporating a more comprehensive review of related literature, the paper could provide readers with a broader understanding of the existing landscape of SLAM research and its implications for UAV applications.

Response 4: Agree. SVSF as a filtering method performs robust results compared with EKF and Particle filter-based methods, especially when facing noise disturbance, and it is worth introducing to complete the filtering methods. Therefore, we expand our filtering methods to include SVSF-based methods, providing readers with a comprehensive understanding. Please find the corresponding revisions highlighted in Subsection Real-time Performance (lines 206-297).

Once again, thank you for your time and careful reviewing. Your comments accurately point out our shortcomings and we think this will make our paper more complete and convincing. We hope our revisions are appropriate.

Reviewer 2 Report

Comments and Suggestions for Authors

1. It would be helpful to specify the scope of the survey more explicitly. For example, does it cover a specific timeframe or particular types of UAV applications?

2. Ensure that all abbreviations, such as UAV (Unmanned Aerial Vehicle) and SLAM (Simultaneous Localization and Mapping), are defined upon first use to aid readers' understanding.

3. It may be beneficial to provide a brief context or background information on why localization and perception are crucial in UAV applications before delving into the technical aspects.

4. Acknowledge any limitations or challenges associated with current visual SLAM techniques, providing a balanced view of their capabilities and constraints.

Comments on the Quality of English Language

Minor editing of English language required

Author Response

Thank you very much for your time and effort to review our manuscript. We appreciate your precious suggestions and please find our responses below.

Comments 1:  It would be helpful to specify the scope of the survey more explicitly. For example, does it cover a specific timeframe or particular types of UAV applications?

Response 1: Agree, explicitly defining the scope of the survey could help readers easily determine whether this survey covers the specific area they are looking for. This survey covered the last 10 years of visual SLAM techniques and aimed at autonomous UAV applications. Therefore, we add our revisions to the Abstract (lines 13-14) to explicitly specify the scope.

Comments 2: Ensure that all abbreviations, such as UAV (Unmanned Aerial Vehicle) and SLAM (Simultaneous Localization and Mapping), are defined upon first use to aid readers' understanding.

Response 2: Agree. We apologize for any confusion caused by the incorrect usage of abbreviations in the survey. We will make sure to thoroughly review and verify all abbreviations during the revision process. Thank you for bringing this to our attention.

Comments 3:  It may be beneficial to provide a brief context or background information on why localization and perception are crucial in UAV applications before delving into the technical aspects.

Response 3:  Agree. The original statement is too short and simple to illustrate why localization and perception are crucial. Thus, we expand it for more detail highlighted in the Introduction (lines 25-30). Please find the corresponding revisions in the re-submitted files, we hope our revisions can make a clear statement.

Comments 4: Acknowledge any limitations or challenges associated with current visual SLAM techniques, providing a balanced view of their capabilities and constraints.

Response 4: Agree. It's important to provide our views of visual SLAM techniques by reviewing related research. However, our manuscript lack of it. Therefore, we revise our manuscript in Section 3 to provide our balanced views in repetitively three aspects (at the end of subsections). discussing its capabilities and constraints, Please find the corresponding revisions highlighted in the re-submitted files.

Once again, thank you for your time and careful reviewing. We hope our revisions are appropriate and correct.

Reviewer 3 Report

Comments and Suggestions for Authors

The paper ''Visual SLAM for Unmanned Aerial Vehicles: Localization and Perception'' is well conceived. The topic chosen by the authors in the paper is up-to-date. In the paper, the authors focused on the presentation of the visual SLAM algorithm applied in UAV applications, with an emphasis on the development of this technique and the observation of the trend of using this technique in UAV applications. The paper presents the development of visual SLAM in three aspects: real-time performance, texture-less environment, and dynamic environment. 

In addition to the introduction and conclusion, the paper consists of four more chapters. In the introduction (Section 1) the authors explained the research problem and design of the paper. In the Section 2 the authors introduced sensor configurations and data processing methods for some preliminaries. In Section 3, the authors reviewed the development of visual SLAM (or odometry) algorithms by features: real-time performance, texture-less environment, and dynamic environment emphasizing state-of-the-art algorithms. In the Section 4 they introduced the widely applied SLAM algorithms in UAVs called visual-inertial SLAM and illustrated how it improves the robustness and accuracy of UAV localization. The learning-based modules for UAV perception were discussed in Section 5 and a comprehensive conclusion was drawn in Section 6 (conclusion).

The cited references are relevant to the research. 

What is worrying about the paper is the lack of classical research, so the paper could be classified as a review paper without classically conducted research (the research design is not appropriate, there are no sections related to the explanation of methods and materials and research results).

In my opinion, this paper does not correspond to the magazine's reputation and should not be published. 

Author Response

Thank you for taking the time to review our manuscript. We appreciate your feedback and the effort. We understand that you believe our manuscript is not suitable for publication. While we respect your professional judgment, we would like to revise my manuscript to address your concerns as best as we can.

We understand your worry about our manuscript lacking classical research since the most of references cover the last 10 years of research. As the response, our revisions are to list and briefly introduce other reviews for covering the classical research. Subsequently, making our manuscript to be more comprehensive and completed. Also, providing readers with a way to learn about the visual SLAM technique through classical research.   

Thank you again for your time and consideration. Please find our revisions highlighted in Section 1 from our re-submitted files. We hope you will take our revisions into consideration. 

Reviewer 4 Report

Comments and Suggestions for Authors

This review focused on the visual SLAM algorithms applied in Unmanned Aerial Vehicles (UAV) applications. This paper summarizes the research, including visual SLAM, visual-inertial SLAM, and learning-based SLAM. The main focus is on three aspects: real-time performance of SLAM algorithms, working in a textureless environment, and working in a dynamic environment. The algorithm components, camera configuration, and data processing methods are also introduced to  give comprehensive preliminaries.

Authors discuss those relative approaches’ problems, advantages, and future trends.

The work is well structured and fully reflects the current development trends of the visual SLAM algorithms. Special attention is paid to Visual-inertial Fusion for UAV Localization. All the algorithms under consideration have a fairly detailed description and links to primary sources.

Author Response

Thank you for your positive feedback and for approving my paper. I truly appreciate your time and effort in reviewing it. I look forward to the opportunity to address any further suggestions or concerns you may have.

Reviewer 5 Report

Comments and Suggestions for Authors

Dear Authors,

Please be advised to the following:

1. All the paper must be checked for English and grammatical mistakes.

2. I think that @Figure 1. General keyframe-based visual SLAM algorithm@ , is not yours , so if you please add the reference!!

3. Figure 4. Different type of maps @ is not cleared and needs enhancment.

4. The conclusion must be enhanced

Comments on the Quality of English Language

All the paper must be checked for English and grammatical mistakes.

Author Response

Thank you very much for your time and effort to review our manuscript. We appreciate your precious suggestions and please find our responses below.

Comments 1:  All the paper must be checked for English and grammatical mistakes.

Response 1: our manuscript indeed has potential English and grammatical mistakes. Thank you for pointing this out. We will check the manuscript thoroughly to make sure correct writing.

Comments 2: I think that @Figure 1. General keyframe-based visual SLAM algorithm@ , is not yours , so if you please add the reference!!

Response 2: Agree. We use Figure 1 to illustrate a general keyframe-based visual SLAM pipeline, however, this figure is actually referred from the classical ORB-SLAM pipeline. we will add the reference down below the figure.

Comments 3:  Figure 4. Different type of maps @ is not cleared and needs enhancment.

Response 3: Agree, we will replace Figure 4 with a clear version, thank you for your careful reviewing.

Comments 4: The conclusion must be enhanced

Response 4: Agree, the conclusion lacks the case studies or examples to support the discussion. we will add some examples to illustrate why UAV applications need robust localization and intelligentized perception. please find the corresponding revisions highlighted in re-submitted files.

Once again, thank you for your time and careful reviewing.  We hope our revisions are appropriate and correct.

Round 2

Reviewer 3 Report

Comments and Suggestions for Authors

The paper ''Visual SLAM for Unmanned Aerial Vehicles: Localization and Perception'' is well conceived.

The authors made corrections in the paper in accordance with the comments of the reviewers.

The paper can be published in this form.

Reviewer 5 Report

Comments and Suggestions for Authors

Dear Authors,

Thanks for your enhanced work